# Radiotherapy Plus Cetuximab for Squamous Cell Carcinoma of the Oral Cavity: A Multicenter Retrospective Study of 79 Patients in Japan

**DOI:** 10.3390/ijerph20054545

**Published:** 2023-03-03

**Authors:** Mitsunobu Otsuru, Souichi Yanamoto, Shin-ichi Yamada, Kohichi Nakashiro, Yosuke Harazono, Tomoyuki Kohgo, Moriyoshi Nakamura, Takeshi Nomura, Atsushi Kasamatsu, Susumu Tanaka, Tadaaki Kirita, Mitomu Kioi, Masaru Ogawa, Masashi Sasaki, Yoshihide Ota, Masahiro Umeda

**Affiliations:** 1Department of Clinical Oral Oncology, Nagasaki University Graduate School of Biomedical Sciences, Nagasaki 852-8588, Japan; 2Department of Oral Oncology, Graduate School of Biomedical and Health Sciences, Hiroshima University, Hiroshima 734-8553, Japan; 3Department of Dentistry and Oral Surgery, Shinshu University School of Medicine, Matsumoto 390-8621, Japan; 4Department of Oral and Maxillofacial Surgery, Faculty of Medicine, University of Toyama, Toyama 930-8555, Japan; 5Department of Oral and Maxillofacial Surgery, Ehime University Graduate School of Medicine, Ehime 791-0295, Japan; 6Department of Maxillofacial Surgery, Graduate School of Medical and Dental Sciences, Tokyo Medical and Dental University, Tokyo 113-8549, Japan; 7Department of Oral and Maxillofacial Surgery, Keiyukai Sapporo Hospital, Sapporo 003-0026, Japan; 8Dental and Oral Medical Center, Kurume University School of Medicine, Kurume 830-0011, Japan; 9Oral Cancer Center, Tokyo Dental College, Chiba 272-8513, Japan; 10Department of Oral Science, Graduate School of Medicine, Chiba University, Chiba 260-8670, Japan; 11The 1st Department of Oral and Maxillofacial Surgery, Graduate School of Dentistry, Osaka University, Osaka 565-0871, Japan; 12Department of Oral and Maxillofacial Surgery, Nara Medical University, Nara 634-8521, Japan; 13Department of Oral and Maxillofacial Surgery, Yokohama City University Graduate School of Medicine, Yokohama 236-0004, Japan; 14Department of Oral and Maxillofacial Surgery and Plastic Surgery, Gunma University Graduate School of Medicine, Maebashi 371-8511, Japan; 15Department of Oral and Maxillofacial Surgery, Tokai University School of Medicine, Isehara 259-1193, Japan

**Keywords:** chemotherapy, multicenter study, oral cancer, prognosis

## Abstract

There are a few reports that focus on radiotherapy (RT) and cetuximab (CET) therapy exclusively for oral cancer. This retrospective study aimed to investigate the efficacy and safety of RT and CET therapy for locally advanced (LA) or recurrent/metastatic (R/M) oral squamous cell carcinoma (OSCC). Seventy-nine patients from 13 hospitals who underwent RT and CET therapy for LA or R/M OSCC between January 2013 and May 2015 were enrolled in the study. Response, overall survival (OS), disease-specific survival (DSS), and adverse events were investigated. The completion rate was 62/79 (78.5%). The response rates in patients with LA and R/M OSCC were 69% and 37.8%, respectively. When only completed cases were examined, the response rates were 72.2% and 62.9%, respectively. The 1- and 2-year OS were 51.5% and 27.8%, respectively (median, 14 months), for patients with LA OSCC, and 41.5% and 11.9% (median, 10 months) for patients with R/M OSCC. The 1- and 2-year DSS were 61.8% and 33.4%, respectively (median, 17 months), for patients with LA OSCC, and 76.6% and 20.4% (median, 12 months) for patients with R/M OSCC. The most common adverse event was oral mucositis (60.8%), followed by dermatitis, acneiform rash, and paronychia. The completion rate was 85.7% in LA patients and 70.3% in R/M patients. The most common reason for noncompletion was an inadequate radiation dose due to worsening general conditions in R/M patients. Although the standard treatment for LA or R/M oral cancer is concomitant RT with high-dose cisplatin (CCRT) and the efficacy of RT and CET therapy for oral cancer is not considered to be as high as that for other head and neck cancers, it was thought that RT and CET therapy could be possible treatments for patients who cannot use high-dose cisplatin.

## 1. Introduction

The standard treatment for oral squamous cell carcinoma (OSCC) is surgery, and if a high-risk factor for recurrences such as postoperative extranodal extension (ENE) or a positive margin is histologically proven, concomitant chemotherapy plus radiotherapy (CCRT) is performed as a postoperative adjuvant therapy. CCRT is also the standard treatment for locoregionally advanced head and neck cancer and recurrent cases with no distant metastases [1].

In 2004, Cooper et al. [2] conducted a randomized controlled trial of radiotherapy (RT) alone vs. RT plus cisplatin (CDDP) for cases of head and neck cancer with histologic evidence of invasion of two or more regional lymph nodes, ENE, and microscopically involved mucosal margins of resection. A randomized controlled trial of high-dose CDDP was performed and reported that concurrent postoperative chemotherapy and RT significantly improved the rates of local and regional control and disease-free survival. In the same year, Bernier et al. [3] also conducted a randomized controlled trial of RT alone vs. RT and CDDP for patients with locally advanced head and neck cancer with pT3 or pT4, nodal stage of 2 or 3, or an unfavorable pathological finding, such as ENE, positive resection margins, perineural involvement, or vascular tumor, and reported that postoperative concurrent administration of high-dose CDDP with RT is more efficacious than RT alone and does not cause an undue number of late complications. Since these results were reported, high-dose CDDP has become widely used as a concomitant drug during RT for patients with locally advanced head and neck cancer or those with a high-risk of recurrence.

On the other hand, in 2010, Bonner et al. [4] conducted a randomized controlled study of RT and RT plus cetuximab (CET) for locoregionally advanced head and neck cancer and reported that CET plus RT significantly improves 5-year overall survival compared with RT alone. In a study of Japanese people, Okano et al. [5] also reported safety and antitumor activity similar to that reported in Bonner et al.’s trial [4]. However, their papers were aimed at patients with oropharyngeal cancer, hypopharyngeal cancer, and laryngeal cancer and did not include cases of OSCC.

Thus, CDDP and CET have been shown to have additional effects when used in combination with RT, and there are some reports on which of the two is more effective. In 2014, Petrelli et al. [6] collected papers, including abstracts comparing RT and CDDP and RT and CET for locally advanced head and neck cancer, and conducted a systematic review and meta-analysis. As a result of an analysis of 1808 cases of 15 papers collected, RT and CDDP were better for both a 2-year overall survival and a 2-year disease-free survival, but 12 of the 15 papers were retrospective studies, and all three prospective studies were phase 2 trials. Out of 903 patients with head and neck cancer in Petrelli et al.’s systematic review [6], excluding conference abstracts, only 66 (7.3%) had OSCC in nine papers [7,8,9,10,11,12,13,14,15]. A review of 13 articles published after Petrelli et al.’s systematic review found that only 428 (5.9%) of 7296 head and neck cancer cases were oral cancers, and there are no studies on RT and CET therapy for oral cancers (Table 1) [16,17,18,19,20,21,22,23,24,25,26,27].

In 2021, Gebre-Medhin et al. [26] reported the results of the first phase III randomized controlled trial of RT and CDDP vs. RT and CET for locoregionally advanced head and neck cancer. The target number of cases was initially set at 618, but RT and CDDP had a significantly higher locoregional control rate, and study inclusion was closed prematurely after an interim analysis in which 298 patients had been randomly assigned. Their study showed that the cumulative incidence of locoregional failures at 3 years was 23%, compared with 9% in the CET versus the CDDP group (*p* = 0.0036), although the 3-year overall survival (OS) was 88% and 78% in the CDDP and CET groups, respectively, which was not significant (*p* = 0.086). They concluded that, regarding locoregional control, CET is inferior to CDDP for concomitant treatment with RT in patients with locoregionally advanced head and neck squamous cell carcinoma, but additional studies are needed to identify possible subgroups that still may benefit from concomitant CET treatment. Notably, only 15 of 291 (5.2%) patients in their study had OSCC.

Based on the results of these studies, many researchers currently consider RT and CDDP as the standard treatments for advanced head and neck cancer. However, in many of the studies mentioned above, the cases were mainly oropharyngeal, hypopharyngeal, and laryngeal cancer, and OSCC cases were either not included or were included in a small number. Many oncologists may feel that CDDP for OSCC is less effective than for pharyngeal cancer. Furthermore, in Vermorken et al.’s EXTREME regimen, a subanalysis showed that CET in combination with platinum-based chemotherapy was more effective in OSCC than other head and neck cancers [28]. These findings suggest that OSCC and other head and neck cancers may have different susceptibilities to drug therapy. Therefore, the Japanese Society of Oral Tumors conducted a preliminary study on the efficacy and safety of RT and CET therapy for OSCC at the facility to which the members belong.

## 2. Materials and Methods

This was a retrospective observational study. The study protocol conformed to the ethical guidelines of the Declaration of Helsinki and the Ethical Guidelines for Medical and Health Research involving Human Subjects by the Ministry of Health, Labor, and Welfare of Japan. Ethics approval was obtained from the institutional review board of Nagasaki University Hospital (14061990-2).

Patients who underwent RT and CET therapy for locoregionally advanced OSCC between January 2013 and May 2015 at the oral and maxillofacial surgery departments of 13 hospitals were enrolled in the study. Cases of histological types other than squamous cell carcinoma or cases of intra-arterial infusion therapy were excluded from the study. The dosing regimen of CET was according to Okano et al.’s paper [5]. Patients received radiotherapy plus weekly doses of cetuximab: 400 mg/m^2^ initial dose, followed by 6 weekly doses at 250 mg/m^2^. Patients who received at least one dose of CET were recruited, even if RT or CET were discontinued.

Patient age, sex, performance status (PS) [29], tumor site, stage, the total dose of RT, completion status of RT, response evaluation criteria in solid tumors (RECIST) [30], and adverse event (CTCAE, ver. 5) [31] were extracted from the medical records. Overall survival (OS) and disease-specific survival (DSS) were calculated using the Kaplan–Meier method.

## 3. Results

### 3.1. Patient Characteristics

Seventy-nine Japanese patients were enrolled in this study. Forty-three patients were males and thirty-six were females, with an average age of 72.5 ± 13.1 years. There were 42 patients with locally advanced (LA) tumors and 37 patients with recurrent or metastatic (R/M) tumors. The most frequent site was the tongue, followed by the mandibular gingiva, buccal mucosa, maxillary gingiva, and the floor of the mouth (Table 2). The sites of recurrence in R/M cases were primary in thirteen, primary and neck in two, primary and distant in one, neck in fourteen, neck and distant in one, and distant in six patients.

### 3.2. Treatment and Response

RT and CET therapy were performed as first-line treatments in 73 patients, while it was performed as a second-line treatment in five, and a conversion treatment in one. The completion rate was 62/79 (78.5%). RT was discontinued at less than 50 Gy in 4 of 42 LA patients (9.5%) and 10 of 37 R/M patients (27.0%). Two out of forty-two LA patients (4.8%) and one out of thirty-seven R/M patients (2.7%) were discontinued after less than three courses of CET. The reasons for discontinuing CET in three patients were side effects such as infusion reaction and interstitial pneumonia, and RT alone was continued in these three patients. The reasons why RT was discontinued was because of the deterioration of the general condition or progressive disease (Table 3).

The response rate was 69% in LA and 37.8% in R/M patients, while the CR rate was 35.7% in LA and 21.6% in R/M patients. When only the completed cases were examined, the response rate was 72.2% in LA and 62.9% in R/M patients (Table 3).

OS for patients with LA was 51.5% at 1 year and 27.8% at 2 years, with a median of 14 months. In R/M patients, the rate was 41.5% at 1 year and 11.9% at 2 years, with a median of 10 months. DSS in LA patients was 61.8% at 1 year and 33.4% at 2 years, with a median of 17 months. In R/M patients, the rate was 76.6% at 1 year and 20.4% at 2 years, with a median of 12 months (Figure 1).

Survival rates were calculated for each treatment period. The 1- and 2-year OS for patients who underwent RT and CET therapy as the first-line treatment were 48.6% and 19.1%, respectively, with a median of 12 months; 1- and 2-year DSS were 55.7% and 26.2%, respectively, with a median of 15 months. When RT and CET were administered as the second-line treatments, the 1- and 2-year OS were both 20%, with a median of 5 months; 1- and 2-year DSS were both 37.5%, with a median of 6 months (Figure 2).

Next, the prognosis was examined according to whether RT and CET therapy had been completed. Among those who completed RT and CET therapy, the 1- and 2-year OS were 50.1% and 22.4%, respectively, with a median of 14 months, and the 1- and 2-year DSS were 58.7% and 30.6%, respectively, with a median of 17 months. In contrast, the 1- and 2-year OS for discontinued patients were 33.3% and 20%, respectively, with a median of 8 months, while the 1-year and 2-year DSS were 39.3% and 23.6%, respectively, with a median of 10 months (Figure 3).

### 3.3. Adverse Events

The most common adverse event was oral mucositis (60.8%), followed by dermatitis, acneiform rash, and paronychia. Infusion reactions were observed in two cases of all grades and one case of grade 4 adverse events, and interstitial pneumonia was observed in seven cases of all grades and five cases of grade 3 or higher adverse events, including one case of grade 5 (Table 4).

## 4. Discussion

This study is the first to examine the efficacy and safety of RT and CET therapy for oral cancer alone. The median OS was 14 and 10 months in LA and R/M patients, respectively. This figure was similar to or slightly lower than previous reports for head and neck cancer.

Bonner et al. [4] reported a median OS in patients with pharyngeal cancer of 49.0 months, and when examined by subsite, the effect in oropharyngeal cancer was the greatest, while that in laryngeal and hypopharyngeal cancer patients was low. This may be due to the high efficacy of cetuximab in oropharyngeal human papillomavirus (HPV)-positive cases. In contrast, the median OS in the present study was worse than in the study by Bonner et al. [4]. Our study was retrospective in nature and included subjects with poor general health conditions and those who did not complete the study. Therefore, although a simple comparison cannot be made, the efficacy of RT and CET therapy for oral cancer is not considered to be as high as that for other head and neck cancers.

In oropharyngeal cancer, HPV-positive cases treated with RT and CET therapy have a favorable prognosis [12,18,21,23,27]. A meta-analysis of epidemiological studies on head and neck cancer in 44 countries, mainly in the United States, Asia, and Europe, reported from 1990 to 2012, showed that the HPV-positive rate in oropharyngeal cancer was 45.8%. The highest positive rate was observed in oropharyngeal cancer, compared with 24.2% in oral cancer and 22.1% in laryngeal cancer [32]. Furthermore, in an epidemiological survey of oral HPV infection in head and neck cancer patients in 29 countries in the United States, Asia, Africa, and Europe, the HPV-positive rate was 24.9% for oropharyngeal cancer and 7.4% for oral cancer. The HPV-positive rate in oropharyngeal cancer was significantly higher than that in oral cancer [33]. In addition, there are many reports that the HPV-positive rate of oral cancer is approximately 4–30% [34,35,36,37,38,39,40] and that of oropharyngeal cancer is approximately 50% [41,42]. This difference in HPV-positive rates may also be reflected in the results of RT and CET therapy. Notably, the HPV-positive rate of oral cancer is reportedly high in Asia [43], because in southeast Asia, there is a habit of chewing tobacco, and HPV infection occurs from wounds on the oral mucosa caused by chewing hard betel nuts [44,45]. However, as there is no custom of chewing tobacco in Japan, this does not appear to be applicable. There is no evidence that RT and CET therapy are more effective for oral cancer in Asia than in other regions. Therefore, there may be factors other than HPV that contribute to the response in the oral cavity.

Based on the results of this study, RT and CET therapy were not as effective for oropharyngeal cancer as for oral cancer among head and neck cancers. In a retrospective study of 4520 patients, Bauml et al. [19] also reported a median OS of 1.5 years, compared with 0.8 years for 320 patients with oral cancer. However, there may be a certain number of cases in which it is effective, as there were long-term survivors who had not observed previous treatments. It is therefore necessary to search for factors specific to such effective cases in the future.

Regarding adverse events, the study by Bonner et al. [4] reported G3-5 mucositis (G3: confluent fibrinous mucositis or may include severe pain requiring narcotic, G4: ulceration, hemorrhage or necrosis, G5: death) in 55.8% of patients, acneiform rash in 16.8%, dermatitis in 35.1%, and infusion reaction in 2.9%, but interstitial pneumonia was not described. Mucositis and dermatitis reported in this study were similar to their report, but acneiform rash was well controlled. However, interstitial pneumonia was found in 6.3% of patients, and G5 interstitial pneumonia was also found in one case. A previous study also reported that interstitial pneumonia developed in 4.5% (9/201) of Japanese patients with head and neck cancer who received cetuximab, and eight out of nine patients had grade 3 or higher [43]. It is necessary to be careful not to generalize these results because there is a possibility that it is unique to Japanese people.

This study has some limitations. First, the number of cases studied was small, focusing only on Japanese patients among Asians, and it is unclear whether they can be generalized. Second, because this was a retrospective study, it is possible that some patients were not suitable for this therapy due to their poor general condition. However, to the best of our knowledge, there have been no studies focused exclusively on oral cancer among Japanese head and neck cancers. In this study, the effect of RT and CET therapy was limited in oral cancers compared with other head and neck cancers. In addition, serious interstitial pneumonia occasionally developed as an adverse event. However, since there were long-term survivors, it may be a promising treatment for patients who cannot receive standard treatment. In addition, if factors specific to effective cases are found, it may become the standard treatment for those patients. In the future, we would like to increase the number of cases and conduct further investigations.

## 5. Conclusions

Although RT and CET therapy seemed to be slightly less effective for oral cancer than for head and neck cancer, it was shown that this therapy could be a safe and effective treatment for RA or R/M OSCC. RT and CET therapy should continue to be considered for cases in which CCRT cannot be used.

## Figures and Tables

**Figure 1 ijerph-20-04545-f001:**
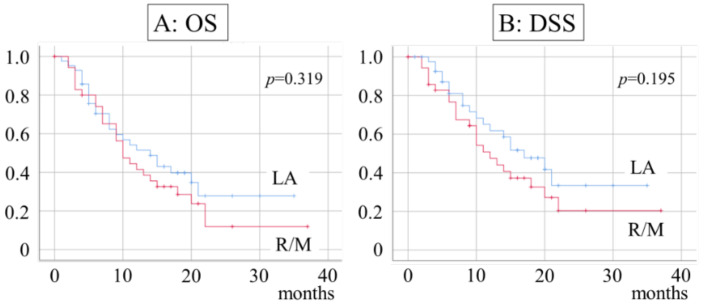
Survival rate and patient background. (**A**) Median OS duration is 14 months in LA and 10 months in R/M patients; (**B**) median DSS duration is 17 months in LA and 12 months in R/M patients.

**Figure 2 ijerph-20-04545-f002:**
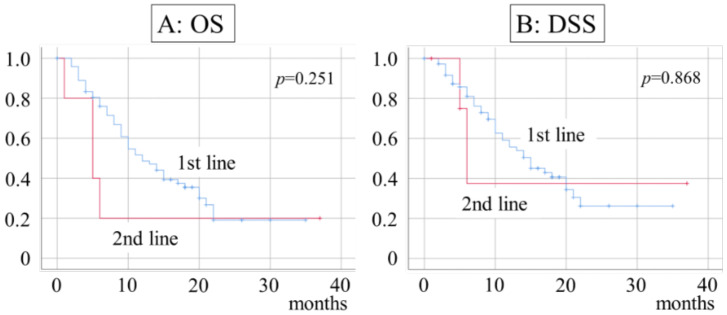
Survival rate and timing of RT and CET therapy. (**A**) Median OS duration is 12 months in first-line and 5 months in second-line patients; (**B**) median DSS duration is 15 months in first-line and 6 months in second-line patients.

**Figure 3 ijerph-20-04545-f003:**
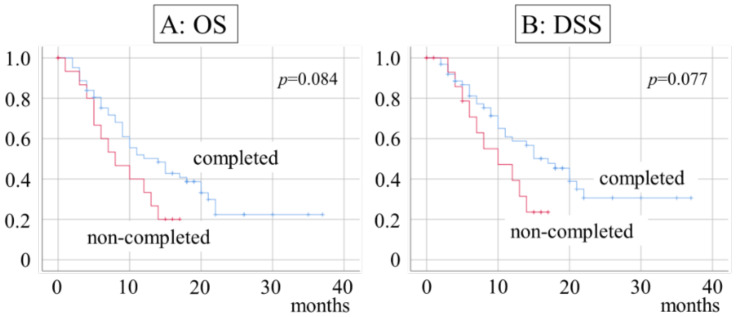
Survival rate and completion of RT and CET therapy. (**A**) Median OS duration is 14 months in completed and 8 months in noncompleted patients; (**B**) median DSS duration is 17 months in completed and 10 months in noncompleted patients.

**Table 1 ijerph-20-04545-t001:** Studies comparing RT and chemotherapy with RT and cetuximab in head and neck cancer.

Author	Year	Study Design	Total Patients	Patients with OSCC	Efficacy of RT and Cetuximab Therapy
(i) Reports in the systematic review by Petrelli [6]					
Caudell [7]	2008	retrospective	132	13 (9.8%)	3-year OS = 75.9%
Jensen [8]	2011	retrospective	76	16 (21%)	median OS = 27.7 months
Beijer [9]	2012	retrospective	125	15 (12%)	median OS = 11 months, 2-year OS = 37%
Lefebvre [10]	2013	phase II	116	0 (0%)	OS = 75% (median follow-up 36 months)
Ley [11]	2013	retrospective	47	3 (6.4%)	3-year OS = 27%
Pajares [12]	2013	retrospective	108	16 (15%)	2-year OS = 89% (HPV positive), 59% (HPV negative)
Huang [13]	2012	retrospective	93	0 (0%)	2-year OS = 58%
Riaz [14]	2014	retrospective	62	0 (0%)	3-year OS = 56.6%
Tang [15]	2014	retrospective	144	3 (2.1%)	3-year OS = 48%
Total			903	66 (7.3%)	
(ii) Reports after the systematic review by Petrelli [6]					
Hu [16]	2014	retrospective	170	52 (30.1%)	3-year OS = 70.9%
Magrini [17]	2016	phase II	70	10 (14.3%)	2-year OS = 68%
Gillison [18]	2019	phase III	805	0 (0%)	5-year OS = 77.9% (HPV positive)
Bauml [19]	2019	retrospective	4520	320 (7.1%)	median OS = 1.5 years (all cases), 0.8 years (oral cancer)
Hamauchi [20]	2019	retrospective	47	2 (4.3%)	median OS = 35.5 months
Jones [21]	2019	phase III	334	0 (0%)	2-year OS = 90% (HPV positive)
Al-Saleh [22]	2019	phase III	40	8 (2.2%)	2-year OS = 57.3%
Mehanna [23]	2019	phase III	334	0 (0%)	2-year OS = 89.4% (HPV positive)
Maddalo [24]	2020	phase II	70	5 (7.1%)	2-year OS = 75.0%
Merlano [25]	2020	phase III	385	14 (3.6%)	TPF→BRT median OS = 45.2 months
Gebre-Medhin [26]	2021	phase III	291	15 (5.2%)	3-year OS = 78.0%
Rischin [27]	2021	phase III	189	0 (0%)	3-year OS = 96.0% (HPV positive)
Total			7296	428 (5.9%)	

Abbreviation: OSCC: oral squamous cell carcinoma; RT: radiotherapy; OS: overall survival; HPV: human papilloma virus; TPF: docetaxel and cisplatin and 5-fluorouracil; BRT: radiotherapy and cetuximab.

**Table 2 ijerph-20-04545-t002:** Patient characteristics.

Factor		LA (*n* = 42)	R/M (*n* = 37)	Total (*n* = 79)
Age (years)	(Mean ± SD)	74.0 ± 11.6	70.7 ± 13.1	72.5 ± 13.1
Sex	Male	26	17	43
	Female	16	20	36
PS	0	25	18	43
	1	16	18	34
	2	1	1	2
Primary site	Tongue	11	20	31
	Mandibular gingiva	11	6	17
	Buccal mucosa	9	4	13
	Maxillary gingiva	5	4	9
	Floor of the mouth	2	2	4
	Others	4	1	5
Stage (at first visit)	1	0	5	5
	2	0	9	9
	3	4	3	7
	4a	28	18	46
	4b	10	0	10
	Unknown	0	2	2

Abbreviation: PS: performance status; LA: locally advanced; R/M: recurrent or metastatic.

**Table 3 ijerph-20-04545-t003:** Treatment and response rate.

Factor		LA (*n* = 42)	R/M (*n* = 37)	Total (*n* = 79)
Line	First	38	36	73
	Second	4	1	5
RT dose (Gy)	Range (median)	16–72 (61)	25–78 (60)	16–78 (60)
Completion rate		36/42 (85.7%)	26/37 (70.3%)	62/79 (78.5%)
Discontinuation at <50 Gy		4	10	14
Discontinuation in less than 3 courses of CET		2	1	3
Response	Complete Response	15 (35.7%)	8 (21.6%)	23 (29.1%)
	Partial Response	14 (33.3%)	6 (16.2%)	20 (25.3%)
	Stable Disease	4 (9.5%)	10 (27.1%)	14 (17.7%)
	Progressive Disease	7 (16.7%)	8 (21.6%)	15 (19.0%)
	Not Evaluable	2 (4.8%)	5 (13.5%)	7 (8.9%)
Overall response rate	%	29/42 (69.0%)	14/37 (37.8%)	43/79 (54.4%)
Response rate in completed cases	%	26/36 (72.2%)	13/26 (50.0%)	39/62 (62.9%)

Abbreviation: RT: radiotherapy; LA: locally advanced; R/M: recurrent or metastatic.

**Table 4 ijerph-20-04545-t004:** Adverse events.

Adverse Events	All Grade	Grade 3–5
Mucositis	48 (60.8%)	21 (26.6%)
Radiation dermatitis	33 (41.8%)	17 (21.5%)
Acneiform rash	34 (43.0%)	3 (3.8%)
Paronychiitis	17 (21.5%)	0 (0%)
Infusion reaction	2 (2.5%)	1 (1.3%)
Interstitial pneumonia	7 (8.9%)	5 (6.3%)

## Data Availability

The datasets used and analyzed during the study are available from the corresponding author upon reasonable request.

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
