# Peer review of "Radiotherapy Plus Cetuximab for Squamous Cell Carcinoma of the Oral Cavity: A Multicenter Retrospective Study of 79 Patients in Japan"

_ijerph, 2023, doi:10.3390/ijerph20054545_

Round 1

Reviewer 1 Report

Comments are in the pdf

Author Response

To Reviewer #1

Line 142 recurrent or metastasis: metastatic tumor have a poor survival rate than those with regional recurrence. So clubbing them together may lead to misleading results. Kindly justify.

(Reply)

Thank you for your advice. In the original paper of RT+CET therapy (Bonner, Lancet Oncol 2010), recurrent and metastatic tumor are collectively described as R/M, and therefore, we also studied those with recurrent and metastatic tumor together as R/M.

Line 222 chewing tobacco: tobacco and betel nut are independent risk factor for cancer right? So even if these cases had HPV, it is not possible to arrive at a conclusive causal inference. Mention this point here for context.

(Reply)

Line 241-243: The description “There is no evidence that RT + CET therapy is more effective for oral cancer in Asia than in other regions. Therefore, there may be factors other than HPV that contribute to response in the oral cavity.” was added.

Line 233 G3-5 mucositis: provide some details in the brackets for contest

(Reply)

Line 251-253: “(G3: confluent fibrinous mucositis or may include severe pain requiring narcotic, G4: ulceration, hemorrhage or necrosis, G5: death)” was inserted after “G3-5 micositis”.

Line 254 Conclusion: Elaborate on the future perspective

(Reply)

The sentence “RT + CET therapy should continue to be considered for cases in which CCRT cannot be used.” was added.

Reviewer 2 Report

ijerph-2258037 Radiotherapy and cetuximab.

General comments:

  1. This manuscript is appropriate for this journal.
  2. The level of English is excellent.
  3. Although I would like a much longer follow-up (5 years), you cite literature that mostly looks at 2-years, so this is appropriate for comparison. A follow-up manuscript at the 5-year mark might be useful

Abstract.

1.     Page 1, line 50:  “RT+CET is a safe and effective treatment for LA or R/M OSCC.” In the Conclusion section, you state this much more carefully (see below). Your research does not determine that it is safe or effective. A more accurate statement would be that the RT+CET for OSCC is less effective than that for oropharyngeal SCC. You can truthfully state that patients who completed the therapy survived longer. I don’t think you can say much about the safety, since many discontinued the therapy because of side effects, and don’t have information if anyone actually died from the therapy (not likely).

2.     You also found that your results generally matched results in those earlier studies (which had a very broad range of results).

3.     I think you need to reword your abstract conclusion.  

Introduction

1.     This is fine.

2.    

Materials and Methods

3.     This is fine.  

Do you have results on how many of the OSCC were HPV-pos? Although the number might be too small to make a statistical analysis, do you have that data, and did you analyze it?

Results

4.     Page 7 line 207: “Therefore, although a simple comparison cannot be made, the efficacy of RT+CET therapy for oral cancer is not considered to be as high as that for other head and neck cancers”. This should be in your abstract and conclusions.

Discussion: My emphasis would be that this therapy is not as effective as for oropharyngeal cancer, comparing it with older studies.

Conclusion: “It was shown that this therapy could be one of the safe and effective treatments for RA or R/M OSCC.” Here you use two qualifying words (“could be” and “one”) that weaken the result, but are more accurate.

Author Response

To Reviewer #2

  1. Page 1, line 50: “RT+CET is a safe and effective treatment for LA or R/M OSCC.” In the Conclusion section, you state this much more carefully (see below). Your research does not determine that it is safe or effective. A more accurate statement would be that the RT+CET for OSCC is less effective than that for oropharyngeal SCC. You can truthfully state that patients who completed the therapy survived longer. I don’t think you can say much about the safety, since many discontinued the therapy because of side effects, and don’t have information if anyone actually died from the therapy (not likely)

(Reply)

Thank you for your valuable advice. The standard therapy for LA or R/M OSCC is concomitant RT with high-dose cisplatin (CCRT), but many patients cannot receive cisplatin due to advanced age, renal dysfunction, or poor general condition. Although the efficacy of RT + CET therapy for oral cancer is not considered to be as high as that for other head and neck cancers, the feasibility of RT + CET therapy is high in comparison with that of CCRT, and the indication of RT+CET therapy should be considered in the future for patients in whom CCRT cannot be performed.

Line 45-50: The next sentences were added.

The completion rate was 85.7% in LA patients and 70.3% in R/M patients. The most common reason for non-completion was inadequate radiation dose due to worsening general condition in R/M patients. Although the standard treatment for LA or R/M oral cancer is concomitant RT with high-dose cisplatin (CCRT) and the efficacy of RT + CET therapy for oral cancer is not considered to be as high as that for other head and neck cancers, it was thought that RT + CET therapy could be the possible treatments for patients who cannot use high-dose cisplatin..

  1. You also found that your results generally matched results in those earlier studies (which had a very broad range of results).

(Reply)

Line 215: “This figure was similar to or slightly lower than previous reports for head and neck cancer.” was added.

  1. I think you need to reword your abstract conclusion.

(Reply)

Line 47-50: I revised the abstract conclusions as follows:

Although the standard treatment for LA or R/M oral cancer is concomitant RT with high-dose cisplatin (CCRT) and the efficacy of RT + CET therapy for oral cancer is not considered to be as high as that for other head and neck cancers, it was thought that RT + CET therapy could be the possible treatments for patients who cannot use high-dose cisplatin.

  1. Do you have results on how many of the OSCC were HPV-pos? Although the number might be too small to make a statistical analysis, do you have that data, and did you analyze it?

(Reply)

We did not investigate HPV-pos.

  1. Page 7 line 207: “Therefore, although a simple comparison cannot be made, the efficacy of RT+CET therapy for oral cancer is not considered to be as high as that for other head and neck cancers”. This should be in your abstract and conclusions.

(Reply) The next sentences were added.

Line 47-50: Although the standard treatment for LA or R/M oral cancer is concomitant RT with high-dose cisplatin (CCRT) and the efficacy of RT + CET therapy for oral cancer is not considered to be as high as that for other head and neck cancers, it was thought that RT + CET therapy could be the possible treatments for patients who cannot use high-dose cisplatin.

Line 276-279: Although RT + CET therapy seemed to be slightly less effective for oral cancer than for head and neck cancer, it was shown that this therapy could be the safe and effective treatments for RA or R/M OSCC. RT + CET therapy should continue to be considered for cases in which CCRT cannot be used.

  1. My emphasis would be that this therapy is not as effective as for oropharyngeal cancer, comparing it with older studies.

(Reply)

Thank you for your valuable comments. We also think that RT + CET therapy for oral cancer is not as effective as for oropharyngeal cancer, just as CCRT is slightly less effective in oral cancer than in oropharyngeal cancer. I added this point to the abstract and the conclusion.

  1. “It was shown that this therapy could be one of the safe and effective treatments for RA or R/M OSCC.” Here you use two qualifying words (“could be” and “one”) that weaken the result, but are more accurate.

(Reply)

Line 277: “one of” was deleted.